# Localization, Disorder, and Entropy in a Coarse-Grained Model of the Amorphous Solid

**DOI:** 10.3390/e23091171

**Published:** 2021-09-06

**Authors:** Premkumar Leishangthem, Faizyab Ahmad, Shankar P. Das

**Affiliations:** School of Physical Sciences, Jawaharlal Nehru University, New Delhi 110067, India; leishangthem@csrc.ac.cn (P.L.); faizya47_sps@jnu.ac.in (F.A.)

**Keywords:** density functional theory, vibrational modes, free energy landscape

## Abstract

We study the role of disorder in producing the metastable states in which the extent of mass localization is intermediate between that of a liquid and a crystal with long-range order. We estimate the corresponding entropy with the coarse-grained description of a many-particle system used in the classical density functional model. We demonstrate that intermediate localization of the particles results in a change of the entropy from what is obtained from a microscopic approach using for sharply localized vibrational modes following a Debye distribution. An additional contribution is included in the density of vibrational states g(ω) to account for this excess entropy. A corresponding peak in g(ω)/$\omega^{2}$ vs. frequency ω matches the characteristic boson peak seen in amorphous solids. In the present work, we also compare the shear modulus for the inhomogeneous solid having localized density profiles with the corresponding elastic response for the uniform liquid in the limit of high frequencies.

## 1. Introduction

In isotropic fluids, the constituent particles move randomly, with the corresponding time-averaged density being constant at all points. This random movement in space is absent at lower temperatures or higher densities, and the crystalline state forms. The latter has a characteristic inhomogeneous density ρ(x) having the symmetry of the corresponding crystal. The particles vibrate around the respective lattice points. Even in computer simulations of a system of hard spheres, it has been seen [1] that a fluid to crystal transition occurs. The hard-sphere fluid transforms into an inhomogeneous state with the particles vibrating around the points of an fcc lattice. This transformation is a result of the competition between the energetic and entropic contributions to the free energy and occurs at the packing $\rho_{0}\sigma^{3}=$ 0.502, which is much lower than the close pack fcc structure with φ= 0.740. The classical density functional theory (DFT) presents a theoretical model [2] for understanding the freezing transition by treating the coarse-grained one-particle density ρ(x) as an order parameter [3]. A typical parametrization of the inhomogeneous density function for the crystalline state involves superposition of Gaussian density profiles centered on a lattice with the long-range order of the crystalline state [4].
(1)$$\rho \left(r\right)=\sum_{i}\phi (|r-R_{i}\left|\right),$$
where the $\left\{R_{i}\right\}$ denotes the underlying lattice. The function ϕ is taken as the isotropic Gaussian $\phi \left(r\right)={\left(\frac{\alpha }{\pi }\right)}^{\frac{3}{2}}e^{-\alpha r^{2}}$. The parameter α characterizes the inverse width or spread of the Gaussian density profile and represents the degree of mass localization in the system. The homogeneous liquid state density is described by Gaussian profiles of very large widths in the limit α→0. Metastable states with aperiodic structures [5,6,7] have also been studied using DFT. They have free energies intermediate between a crystal and a homogeneous liquid state. These metastable states consist of localized density profiles around the respective points $\left\{R_{i}\right\}$ of an amorphous lattice. Such local minima of the free energy functional exist for hard-sphere interactions and hard-core Hertzian potentials. The thermodynamic properties of the inhomogeneous amorphous solid are computed, assuming the system to be in a single phase. The amorphous nature of the underlying structures plays a crucial role in producing the metastable free-energy minima. Breaking of the isotropic symmetry of the liquid state leads to the development of the transverse sound modes in the crystal which are of the Goldstone modes [8]. Transverse sound modes occur in the amorphous solid state as well. However, the vibrational states of glassy materials are beyond the simple plane-wave phonon picture for crystals.

The crystalline and the amorphous (termed as glass) states of matter have a characteristic high degree of mass localization compared to that of the liquid state. In the present context of DFT, this refers to free-energy minima for the system corresponding nonzero values of the parameter α. The localized particles in the disordered glassy state are on a metastable amorphous lattice structure without any long-range order. The individual particles vibrate around the sites, which remain localized on a disordered lattice over long times. For the amorphous solid, density of vibrational modes is modified from the predictions of the Debye distribution. The excess in the density of states (DOS) g(ω) for a disordered system is at the THz frequency (ω) range and appears as a peak in the reduced DOS representation as in $g\left(\omega \right)/\omega^{2}$ vs. ω plot. This peak is analogous to the so-called boson peak, whose height decreases and location shifts towards the higher frequencies with the increase of pressure or density. Inelastic scattering studies of light and neutron from amorphous solids [9,10] has established the presence of the boson peak.

Several authors have studied various models for understanding the boson peak formation. These include analysis based on (a) potential energy landscape description to describe a transition from a saddle-point-dominated phase without phonons to a minima-dominated phase with low-energy phonons [11], (b) localized phonons in cohesive clusters [12,13], (c) an-harmonic interaction potentials [14] for a cluster of particles, (d) domain walls between configurations forming a mosaic of disordered states [15], (e) descriptions in terms of crystalline lattices of springs with randomly assigned stiffness [16], (f) randomly fluctuating elastic constants [17,18], and (g) the disordered network solids with fixed bond connectivity [19]. The model for a disordered solid is studied [16] in terms of a geometrically perfect crystal with random interactions between nearest neighbors or in terms of a crystal having particles with different masses. A common aspect of the boson peak models is that they are based on the existence of localized modes in the amorphous solid [20] and are a manifestation of the disorder. The system’s free energy reaches a minimum in the equilibrium state with contributions from the vibrational modes in the solid.

In the present work, we demonstrate the role of disorder in producing the metastable states with a degree of mass localization, intermediate between a liquid and a crystal with long-range order. Using coarse-grained description in terms of the density field, we obtain that localization of the particle over intermediate length scales makes a change in the system’s entropy. This modified entropy is obtained by including the excess density of states in the form of a boson peak. For the inhomogeneous states using the standard formulation of DFT, thermodynamic properties like pressure and elastic constants like shear modulus are also obtained. For this, we compute the first and second derivatives of the free energy with respect to the average density. In the present work, we also consider how the solid-like properties, like the shear modulus for the inhomogeneous disordered states (with mass localization), change compared to the similar property for the uniform liquid in the high-frequency limit. We organize the paper as follows. In Section 2, we discuss the calculation of entropy of the hard-sphere system using a continuum model in terms of the coarse-grained density. We consider a model for the solid-like system in terms of vibrational modes and a choice of the appropriate density of states g(ω). In the next section, we discuss the calculation of the total free energy taking into account the role of the interaction and identify the amorphous metastable states with inhomogeneous density ρ(x). We analyze the appropriate density of states for the solid-like state with a low degree of mass localization (compared to a crystal) and identify the characteristic boson peak. In Section 4, we calculate the elastic constants for the amorphous solid-like state by extending the DFT to include the weighted density approximation. The paper ends with a discussion of the key points.

## 2. Entropy of Delocalization

### 2.1. The Coarse Grained Model

Let us consider a system of *N* identical particles of mass *m* and the position and momentum coordinates $\{r_{\alpha },p_{\alpha }\},$ for α=1,..N. The whole set of phase space variables are to be denoted as $\left\{r^{N},p^{N}\right\}$. At the microscopic level, the one particle density $\hat{\rho }\left(x\right)$ is formally defined as a function of the phase space variables $\left(r^{N},p^{N}\right)$,
(2)$$\hat{\rho }\left(x\right)=\sum_{\alpha =1}^{N}\delta \left(x-r_{\alpha }\right).$$
The microscopic Hamiltonian for the system of *N* particles is given by
(3)$$H_{N}=\sum_{\alpha =1}^{N}\frac{p_{\alpha }^{2}}{2m}+U+U_{\mathrm{ext}}\equiv H_{N}^{0}+U_{\mathrm{ext}}\;.$$
where $H_{N}^{0}$ is the intrinsic part of the Hamiltonian *N* particle system, and $U(r_{1},\ldots .,r_{N})$ is the interaction energy of the *N* particle system. $U_{\mathrm{ext}}$ is the external field contribution to the Hamiltonian and is obtained as an interaction term with a local field arising from a one-body potential ϕ,
(4)$$U_{\mathrm{ext}}=\sum_{\alpha }\phi \left(r_{\alpha }\right)=\int dx\phi \left(x\right)\hat{\rho }\left(x\right)\;.$$
The coarse-grained density function corresponding to the distribution function *f* is then obtained as $\rho \left(x\right)=\mathrm{Tr}\hat{\rho }\left(x\right)f$, where the operation “Tr” refers to statistical mechanical average. For the equilibrium density, we choose the standard grand canonical ensemble.

The free energy of the liquid is obtained as a functional of the coarse-grained density function ρ(x) as a sum of two parts—the ideal gas term and the interaction term:(5)$$F\left[\rho \right]=F_{\mathrm{id}}\left[\rho \right]+F_{\mathrm{ex}}\left[\rho \right].$$
The ideal gas part $F_{\mathrm{id}}$ is obtained exactly by setting the interaction potential U=0 in the Hamiltonian. In this case, Hamiltonian $H_{N}^{o}$ of the *N* particle system only has the kinetic energy term $K=\sum_{\alpha }p_{\alpha }^{2}/2m$ and thus explicitly integrating out the 3N momentum variables, the grand canonical partition function Ξ is obtained as
(6)$$\Xi =\mathrm{Tr}\exp\left[-\beta \left\{H_{o}-\sum_{\alpha }u\left(r_{\alpha }\right)\right\}\right]=\exp\left[\Lambda_{0}^{-3}\int dxe^{\beta u(x)}\right]$$
where u(x)=μ−ϕ(x), in terms of the external field ϕ and $\Lambda_{0}=h/\sqrt{2\pi mk_{B}T}$, is the thermal de Broglie wavelength for the liquid particles. Using the expression (Equation 6), the coarse grained density is obtained for noninteracting case in the exact form,
(7)$$\rho \left(x\right)=\frac{\exp[\beta u(x)]}{\Lambda_{0}^{3}},$$
where $\beta u\left(x\right)=\ln[\Lambda_{0}^{3}\rho \left(x\right)]$. The free energy functional F[ρ(x)] for the noninteracting system is obtained in terms of the coarse grained density by generalization of the corresponding thermodynamic relation,
(8)$$F_{\mathrm{id}}\left[\rho \right]=\mu \int dx\rho \left(x\right)-\int dx\phi \left(x\right)\rho \left(x\right)-\beta^{-1}\int \rho \left(x\right)dx\;$$
We have used the ideal gas equation of state in writing the second term on the right-hand side. Therefore, using the expression (Equation 7) we obtain
(9)$$F_{\mathrm{id}}=\beta^{-1}\int dx\rho \left(x\right)\left[\ln\left(\Lambda_{0}^{3}\rho \left(x\right)\right)-1\right].$$
For the noninteracting system of *N* particle, the partition function is $Z_{N}={\left(V/\Lambda_{0}^{3}\right)}^{N}/N!$, and thus the ideal gas free energy is obtained from the logarithm of $Z_{N}$ for the equilibrium state. Therefore, $\beta F_{\mathrm{id}}=V\rho_{0}\left(\ln\left(\rho_{0}\Lambda_{0}^{3}\right)-1\right)$ for the uniform density fluid $\rho_{0}$, which is the limit $\rho \left(x\right)\to \rho_{0}$ in Equation (Equation 9). From Equation (Equation 9), it is easy to see that the entropy drops upon localization of the particles due to the restriction of available phase space. The ideal gas contribution for the *N* particles system changes by an amount $\Delta F_{\mathrm{id}}$ when $\rho_{0}\to \rho \left(x\right)$:(10)$$\Delta F_{\mathrm{id}}=\rho_{0}\int dr\tilde{\rho }\left(r\right)\left[\ln\tilde{\rho }\left(r\right)\right],$$
where $\tilde{\rho }\left(r\right)=\rho \left(r\right)/\rho_{0}$. As by definition the quantity $\tilde{\rho }\left(r\right)$ is always positive, using the Gibbs inequality {xlnx−x+1}≥0 for positive *x*, it follows that $\Delta F_{\mathrm{id}}\geq 0$. The relation (Equation 7) between the chemical potential and the density is inverted here, giving an exact expression for the free energy functional. In this regard, it is essential to note that the corresponding free energy for the system, expressed as a function of the density ρ(x), is exact. It is, however, possible only for the so-called ideal-gas contribution for the noninteracting system. The role of interactions are accounted for by introducing the direct correlation functions $c^{\left(i\right)}$’s for *i* = 1, 2, 3, which are defined as successive order functional derivative of an excess contribution $F_{\mathrm{ex}}$ to the free energy due to interactions. We will be discussing the interacting system in the next section.

Two length scales are inherent in the coarse-grained picture for the solid presented above. The interaction potential between the constituent particles has the characteristic length scale, which is the hard-sphere diameter σ in the present case. The other length is $\ell =1/\sqrt{\alpha }$ used in parametrization of the inhomogeneous density function (Equation 1). In the coarse-grained description, the scale *ℓ* signifies the degree of localization of mass in the system, and the limit $\alpha \sigma^{2}\to 0$ or ℓ>>σ represents density profiles of uniform liquid with an average $\rho_{0}=N/V$. In this case, the sum in the RHS of Equation (Equation 1) has matching contributions from a large number of terms, each of which corresponds to a Gaussian profile centered on a corresponding site $\in \{R_{i}\}$. On the other hand, the limit $\alpha \sigma^{2}>>1$ or ℓ<<σ corresponds to sharply localized density profiles like that in a crystal. The ratio ℓ/σ is generally referred to as the Lindeman parameter for the crystal. For $\alpha \sigma^{2}\;\geq \;50$, the density ρ(r) is constructed in terms of the contributions from non-overlapping density profiles and Equation (Equation 9) is well approximated by its asymptotic value for large α,
(11)$$f_{\mathrm{id}}\left[\rho \right]\approx -\frac{5}{2}+\ln\left[\Lambda_{0}^{3}{\left(\frac{\alpha }{\pi }\right)}^{\frac{3}{2}}\right].$$
For lower values of α where the overlapping of the Gaussian profiles from different sites are relevant in the sum in Equation (Equation 1), $f_{\mathrm{id}}$ is evaluated numerically from the integral
(12)$$f_{\mathrm{id}}\left[\rho \right]=\int dr\phi \left(r\right)\left\{\ln\left[\Lambda_{0}^{3}\int dR\phi \left(r-R\right)\left(\delta \left(R\right)+\rho_{o}w\left(R\right)\right)\right]-1\right\}\;\;.$$
The free energy in the right-hand side of (Equation 12) is averaged over a random set of Gaussian centers $\left\{R_{i}\right\}$ for the density profiles, and is obtained in terms of the site–site correlation function w(R). However, it is helpful to note here that even if the Gaussian centers are on a regular crystalline lattice, the result for the non-interacting system is not changed. It follows directly by expressing the inhomogeneous density function ρ(r) in terms of reciprocal lattice vector (RLV) expansion:(13)$$\rho \left(r\right)=\rho_{0}+\rho_{s}\sum_{i}\zeta_{i}e^{iG_{i}\cdot r},$$
where $\left\{G_{i}\right\}$ denotes the RLV [21], and for the fcc structure the solid density $\rho_{s}=4/a_{0}^{3}$ in terms of the lattice constant $a_{0}$. The RLV expansion for the density reduces to the form (Equation 1) when we identify $\zeta_{i}=\exp\left[-G_{i}^{2}/\left(4\alpha \right)\right]$. For smaller values of α, the sum in the expansion for density has to include contributions from increasingly the larger number of RLVs.

Figure 1 shows the variation of the total entropy *S* (in units of $k_{B}$) obtained by evaluating the ideal gas part of the free energy $f_{\mathrm{id}}$ from the expression (Equation 9) using two different approximations for the inhomogeneous density ρ(x). First, in Equation (Equation 1), ρ is described with in terms of Gaussian density profiles centered on a random lattice $\left\{R_{i}\right\}$ in real space. Second, in Equation (Equation 13), density is expressed in terms of an expansion in reciprocal wave vectors {G} corresponding to a regular lattice with long range order. We use the fcc structure going up to Gσ=14. The ρ(x) in the respective cases are used in expression (Equation 9) to obtain the $f_{\mathrm{id}}$. These results shown in the main panel correspond to a hard-sphere system with packing-fraction $\varphi =\pi \rho_{0}\sigma^{3}/6=0.576$. The thermal wavelength is kept a constant at Λ=0.025σ. For a crossover value of $\alpha \leq \alpha_{0}$, the entropic contribution is evaluated numerically from the integral given in Equation (Equation 12). The pair function w(R) is approximated here with pair function for the Bernal’s random structure [22] and is generated through Bennett’s algorithm [23]. We have obtained w(R) using the following relation:(14)$$w\left(R\right)=g_{B}\left[R{\left(\varphi /\varphi_{0}\right)}^{\frac{1}{3}}\right],$$
where φ denotes the average packing fraction and $\varphi_{0}$ is a scaling parameter for the structure [24] such that at $\varphi =\varphi_{0}$, Bernal’s structure is obtained. The important thing to note here that for $\alpha <\alpha_{0}$, the free energy curve (shown as the solid line) deviates completely from the asymptotic result. This line extends to the correct α→0 limit $\left(\ln\left\{\rho_{0}\Lambda^{3}\right\}-1\right)$ corresponding to the uniform liquid state. In the DFT model with coarse-grained density ρ(x) expressed in terms of Gaussian profiles of inverse width α, a corresponding $\alpha_{0}$ is identified such that for all $\alpha <\alpha_{0}$ the asymptotic formula (Equation 11) *deviates* from the entropy curve (shown in Figure 1) of the coarse-grained model. The inset of Figure 1 shows the weak dependence of $\alpha_{0}$ on density $\rho_{0}$.

Note, here, that the modified entropic contribution discussed above, corresponding to small values of α or more spread out density distributions, be it around a random structure or a regular set of lattice points, it is a manifestation of the delocalization of the particles and in the α→0 it goes to the case of an isotropic liquid.

### 2.2. Microscopic Description

As an alternative to the coarse-grained model, evaluating the corresponding partition function (for a microscopic model) obtains the free energy for the system. The microscopic description is introduced here to focus on the density of vibrational modes in the amorphous state with inhomogeneous mass distribution. The description involving the vibrational modes is needed to study the corresponding modifications of the density of states in the metastable liquid. Though no long-range symmetry breaking occurs, the glassy state still supports transverse sound waves. These transverse modes are described in terms of vibrational excitations over an amorphous structure constituting the phonon modes. The respective free energies obtained from the continuum and microscopic approaches closely agree in the form of parameters of the (density) field-theoretic model that are identified with the density of states and characteristic cut-off frequency of the vibrational modes. The set of vibrational modes for the density of states g(ω)dω between frequencies ω and ω+dω contributes to this. For an average $\rho_{0}$ number of particles per unit volume, we have a constraint on *g* maintaining the total number of vibrational modes in three dimensions,
(15)$$\int_{0}^{\omega_{m}}g\left(\omega \right)d\omega =3\rho_{0}\;\;.$$
The cut-off frequency $\omega_{m}$ is a characteristic property of the material and sets the shortest time scale for the vibrational modes. For fixed $\rho_{0}$, the upper cut-off $\omega_{m}\left[g\right]$ in the integral in Equation (Equation 15) is treated as a functional of the density of states g(ω). In the constant NVT ensemble, the partition function for 3N harmonic oscillators is
(16)$$Z=\prod_{i=1}^{3N}\sum_{n_{i}=0}^{\infty }\exp\left[\frac{-\hbar \omega_{i}}{k_{B}T}\left(n_{i}+\frac{1}{2}\right)\right],$$
where $\omega_{i}$ is the frequency of the *i*-th vibrational mode in the system. The free energy of the non-interacting system is obtained in units of $\beta^{-1}$ as
(17)$$\hat{f}\left[\bar{g}\right]\;\equiv \;\frac{\bar{E}-TS}{V}=\int_{0}^{1}\kappa \left(\varepsilon_{m}x\right)\bar{g}\left(x\right)dx\;,$$
where $\bar{E}$ denotes the average kinetic energy of the *N* particles and scaled energy $\varepsilon_{m}=\beta \hbar \omega_{m}$. We write the free energy as $\hat{f}$ with a hat to indicate that it is obtained from microscopic considerations. The density of states g(ω) is written in Equation (Equation 17) in a scaled form $\bar{g}\left(x\right)$, as a function of the variable $x=\omega /\omega_{m}$, i.e., $\bar{g}\left(x\right)={\left(3\rho_{0}\right)}^{-1}\omega_{m}g\left(\omega \right)$. In this case, $\bar{g}\left(x\right)$ is only nonzero for 0≤x≤1. The function κ(y) in the right hand side of Equation (Equation 17) is obtained as
(18)$$\kappa \left(y\right)=-\frac{1}{4}\left\{y+\ln\left[1-e^{-y}\right]-2y/\left(e^{y}-1\right)\right\}.$$
Please see Appendix A for details of obtaining the above result for the function κ(y). The function $\kappa (\varepsilon_{m}x)$ determines the functional dependence of $\hat{f}\left[g\right]$ on the scaled density of states $\bar{g}$ as given in Equation (Equation 17). A link between the density of states g(ω) and the mass localization parameter α, is obtained by equating the results (Equation 9) and (Equation 17) for the ideal gas part of the free energy following respectively from the coarse-grained model and the microscopic calculation.

#### 2.2.1. Debye Distribution

If the density of states is the Debye distribution, we obtain
(19)$$g\left(\omega \right)\equiv g_{D}\left(\omega \right)\equiv \frac{9\rho_{0}}{\omega_{D}^{3}}\omega^{2},\;\;$$
where the upper cut-off $\omega_{m}$ is the Debye frequency $\omega_{D}={\left(6\pi \rho_{0}\right)}^{1/3}c$, in terms of the speed of sound *c*. The corresponding scaled distribution ${\bar{g}}_{D}$ is obtained as a function of the scaled variable ${\bar{g}}_{D}\left(x\right)\;=\;3x^{2}$ where $x\;=\;\omega /\omega_{D}$. In terms of the reduced variables, we obtain $\varepsilon_{m}\equiv \varepsilon_{D}=\beta \hbar \omega_{D}$. In the limit, $\varepsilon_{D}\;<<\;1$, when the density of states g(ω) is the Debye distribution $g\equiv g_{D}$ in Equation (Equation 17), we have to the leading orders in $\varepsilon_{D}$:(20)$$\hat{f}\left[{\bar{g}}_{D}\right]=-\frac{5}{2}+3\ln\varepsilon_{D}+O\left(\varepsilon_{D}^{2}\right).$$
The result (Equation 20) for the free energy per particle calculated from the partition function of the microscopic model is identical to the asymptotic formula (Equation 11) where we obtain the (reduced) Debye frequency as
(21)$$\beta \hbar \omega_{D}\equiv \varepsilon_{D}=\sqrt{\frac{\alpha }{\pi }}\Lambda .$$
By comparing numerical evaluation of the integral on the right hand side of Equation (Equation 17) (for the choice $\bar{g}\equiv {\bar{g}}_{D}$), with the asymptotic result (Equation 20) for different values of $\varepsilon_{D}$, a maximum value $\varepsilon_{D}^{max}$ of $\varepsilon_{D}$ is identified. It is $\varepsilon_{D}$ up to which the two results agree. Figure 2 shows this comparison. The point at which the two curves (approximate and exact) as given by Equations (Equation 20) and (Equation 17), respectively, deviate is marked with an arrow at $\varepsilon_{D}^{max}=2.5$ in the main panel. In the inset of the same figure, we show the corresponding agreement between the two formulas for entropy (calculated from DFT) shown in Figure 1. The arrow shown in this inset corresponds to $\varepsilon_{D}=$ 0.069, i.e., using the relation (Equation 21), it follows that (for Λ/σ= 0.025) the corresponding $\alpha \;=\;24\;\approx \;\alpha_{0}$. From this, it is clear that over the entire range of α values in the DFT model, the Formula (Equation 11) is in agreement with the results of the corresponding microscopic model with a Debye density of states. However, as the actual entropy, *S*, deviates from the asymptotic Formula (Equation 11) for $\alpha <\alpha_{0}$, the corresponding density of states g(ω) in the microscopic model gets modified from the Debye form.

#### 2.2.2. Excess Density of States

To summarize the previous section, the free energy curve using DFT for $\alpha \;\geq \;\alpha_{0}$ follows the asymptotic form (Equation 11), and this is identical with the form (Equation 20) obtained in the microscopic framework using the Debye distribution $g_{D}\left(\omega \right)$ as density of states g(ω). However, for $\alpha \;<\;\alpha_{0}$, the departure of the entropy curve from the asymptotic form (Equation 11) as shown in Figure 1 requires modifying in the microscopic model the corresponding density of states for g(ω) from the Debye form $g_{D}\left(\omega \right)$. The normalization condition (Equation 15) for the modified g(ω) is maintained with a corresponding upper cut-off in frequency $\omega_{m}$ or equivalently $\varepsilon_{m}$ (=$\beta \hbar \omega_{m}$) which is different from that of the Debye distribution. The upper cut-off $\omega_{m}$ (and thus $\varepsilon_{m}$) will depend on the form of function g(ω) as well as the value of width parameter α (<$\alpha_{0}$), as the corresponding entropy, calculated in the coarse grained model has to match with the microscopic results (Equation 17)–(Equation 18) with the modified g(ω). We require that the function $\varepsilon_{m}\left(\alpha \right)$ → $\varepsilon_{D}\left(\alpha \right)$ (of Equation (Equation 21)) as $\alpha \to \alpha_{0}$. The correction part in this functional relation is denoted by C(α), such that $\varepsilon_{m}\left(\alpha \right)=\varepsilon_{D}\left(\alpha_{0}\right)+C\left(\alpha \right)$ with C(α)→0 as $\alpha \to \alpha_{0}$. For the scaled distribution function $\bar{g}\left(x\right)$ we make a modification over the corresponding Debye form ${\bar{g}}_{D}\left(x\right)=3x^{2}$ in terms of the function Δ, such that $\bar{g}\left(x,x_{m}\left(\alpha \right)\right)=3x^{2}\left(1+\Delta \left(x,\alpha \right)\right)$. As $\alpha \to \alpha_{0}$, we must have $g\to {\bar{g}}_{D}$ for all frequencies *x*, and we express Δ with the separation of variables: $\Delta \left(x,\alpha \right)=B\left(\alpha \right)\tilde{\Delta }\left(x\right)$, where B(α)→0 as $\alpha \to \alpha_{0}$. In order to maintain the positivity of the density of states, we also require that the function $\tilde{\Delta }\left(x\right)$ is in the range ±1. The *x* dependence of the $\tilde{\Delta }\left(x\right)$ is chosen to have an intermediate peak over the whole frequency range, and the position of the peak of $\tilde{\Delta }\left(x\right)$ is kept fixed at x= 0.22. In Figure 3, we show the results obtained for the appropriate density of states $\bar{g}\left(x\right)$ which reproduce the $f_{\mathrm{id}}$ for $\alpha <\alpha_{0}$. We study a hard sphere system at a fixed density φ = 0.576. The reduced density of states $g\left(\omega \right)/\omega^{2}$ vs. $\omega /\omega_{D}$ is shown in Figure 3 for five different values of the localization parameter $\ell =1/\sqrt{\alpha }$. The variation of the corresponding upper cutoff $\omega_{m}$ with width parameter α is shown in the main panel of Figure 4. The peak frequency $\omega_{m}$ is smaller than the Debye frequency $\omega_{D}$ for $\alpha <\alpha_{0}$ and approaches $\omega_{D}$ as $\alpha \to \alpha_{0}$.

## 3. The Free Energy Landscape

In the density functional theory, for identification of the equilibrium state of the fluid, a proper thermodynamic potential or free energy is minimized with respect to the density ρ(x). Free energy is expressed as a sum of two parts which are, respectively, the free energy of the non-interacting system $F_{\mathrm{id}}$ and the contribution $F_{\mathrm{ex}}$ due to interactions between the particles. The part $F_{\mathrm{id}}$ we have already discussed in details above. The interaction part $F_{\mathrm{ex}}$ of the free energy is obtained in a functional Taylor series expansion around the corresponding free energy for the uniform density state in terms of the density fluctuation $\delta \rho \left(x\right)\;=\;\rho \left(x\right)\;-\;\rho_{0}$. For the *N* particle system we obtain, the Ramakrishnan–Yussouff (RY) functional,
(22)$${\Delta F}_{\mathrm{ex}}=-\frac{1}{2}\int dx_{1}\int dx_{2}c(|dx_{1}-x_{2}\left|;\rho_{0}\right)\delta \rho \left(x_{1}\right)\delta \rho \left(x_{2}\right)\;\;.$$

At low densities, the state with spatially uniform mass distribution has the lowest free energy. However, at higher densities, the crystalline state with highly localized density profiles centered around the points of a lattice with long-range order has lower free energy and represents the equilibrium state. Metastable minima corresponding to the above free energy functional, intermediate between the isotropic liquid and the crystalline state, have been identified in several extensions of traditional DFT methods [5,6]. Those metastable states have characteristic ρ(r) represented in terms of Gaussian profiles centered on the points of an amorphous lattice. However, the inverse width parameter α for these metastable states is generally lower than those for the sharply peaked density profiles depicting the crystalline state. A qualitatively different free energy minimum, with the optimum density function characterized by much smaller α values, has also been obtained subsequently [7]. The low α minimum appears at α≈18 for $\rho_{0}=1.12$. We have chosen here to describe the amorphous lattice $\left\{R_{i}\right\}$ in terms of the Bernel structure. However, metastable minima with the less localized density profiles also occur for other choices of the random structure $\left\{R_{i}\right\}$ [25,26]. The expansion (Equation 22) for the free energy of the liquid works as a better approximation when the coarse-grained density corresponds to a low degree of mass localization compared to that for the crystalline state with sharp density profiles.

The role of the interaction term enters here through the direct correlation function c(r). The local minima of the free energy, signifying a low degree of mass localization, have been obtained for various interaction potentials. These include hard sphere interaction [7,27], Lennard–Jones [28], soft-sphere interaction [29], and Hertzian potentials [30]. A typical result obtained for the hard-sphere system in which the solution of the Percus–Yevick equation [31] has been used to compute the direct correlation function c(r), is displayed in Figure 5. In this case, minimization of the total free energy shows that disorder plays a vital role in producing (in the free energy landscape) the local minima, which signify the metastable states [32] in the partially delocalized region $\left(\alpha \leq \alpha_{0}\right)$. The modified entropy is matched with changed density of vibrational states, and is apparent from the fact that the interaction part of the free energy $F_{\mathrm{ex}}$ is very different in the respective cases of ordered and disordered structures. If the Gaussian profiles for ρ(r) have their respective centers on a lattice with long-range order similar to the crystalline state, *no metastable minimum for the total free energy is obtained* in the small α ($<\alpha_{0}$) region. Figure 6 shows this case with no minima appearing in the low α region. These low α or partially localized state exists only if $F_{\mathrm{ex}}$ is computed with the lattice points $\left\{R_{i}\right\}$ on a random structure. Thus, disorder is essential for the metastable minimum of the free energy in the delocalized region ($\alpha \leq \alpha_{0}$). For disordered systems with weaker mass localization than the crystal, the corresponding density of states g(ω) differs from the Debye distribution. This modified density (over Debye) produces the entropy contribution appropriate for the delocalized amorphous state. This density of states is consistent with a boson peak seen in amorphous solids.

For a specific interaction potential, density (φ) dependence of properties like peak-height ($B_{h}$) and peak-position ($\omega_{p}$) of g(ω) is determined in terms of the optimum $\alpha_{\min}$. In Figure 7, we show for the hard sphere potential, the dependence of $\alpha_{\min}$ on the packing fraction φ. Using this dependence, we obtain the corresponding boson peak curve (see Figure 3, for example). In each case, the properties $B_{h}$ and $\omega_{p}$ for the distribution curve at the $\alpha_{\min}$ corresponding to packing fraction φ are obtained. Linking of $\alpha_{\min}$ to φ is done using Figure 7. The density (φ) dependence of height and position of the boson peak are displayed, respectively, in Figure 8 and Figure 9. Insets of Figure 8 and Figure 9, respectively, display the corresponding results from experiments. The trends seen from the experimental data are the same as the theoretical model shown in the respective main panels. With the increase of density, the boson peak height decreases, and the peak shifts to higher frequencies. Thus accounting for the entropy for delocalized states by modifying (in the microscopic model) the corresponding density of states in the form of boson peak agree with the experimental data for amorphous metastable systems. Further studies with various interaction potentials are needed to better understand this link between microscopic and coarse-grained models.

## 4. Elasticity of the Localized State

The elastic constants for the amorphous metastable state is an important property to understand its solid-like nature. In the DFT, equilibrium free energy of the inhomogeneous state is obtained as a function of the average density $\rho_{0}$. The bulk modulus *K* of the isotropic solid is obtained in terms of the second derivatives of the free energy [24] with respect to $\rho_{0}$.
(23)$$K=\rho_{0}^{2}\frac{\partial^{2}f}{\partial \rho_{0}^{2}}\;\;.$$
The pressure in the solid is also obtained from the first derivative of the free energy as
(24)$$P=\rho_{0}\frac{\partial f}{\partial \rho_{0}}-f.$$
We use the modified weighted density functional approximation (MWDA) [34] for calculating the free energy of the inhomogeneous liquid in the metastable state. This is an effective medium approach in which the nonuniform solid is mapped to an equivalent homogeneous liquid of lower density. In calculating the excess part per particle $f_{\mathrm{ex}}=F_{\mathrm{ex}}/N$ using MWDA [35,36,37,38] in the canonical ensemble, a self-consistent integral equation [34] is obtained for the density of the effective liquid $\hat{\rho }$. The corresponding packing fraction $\hat{\varphi }=\pi \hat{\rho }\sigma^{3}/6$ in terms of the suitably chosen free energy function $f_{\mathrm{ex}}\left(\hat{\varphi }\right)$.
(25)$$\hat{\varphi }=\frac{I(\hat{\varphi },\alpha )}{2f_{\mathrm{ex}}^{\prime }\left(\hat{\varphi }\right)+\varphi f_{\mathrm{ex}}^{\prime \prime }\left(\hat{\varphi }\right)}$$
where the integral I is defined as
(26)$$I=N^{-1}\int dx\int dx^{\prime }\rho \left(x\right)\rho \left(x^{\prime }\right)c(|x-x^{\prime }\left|;\hat{\varphi }\right)\;\;\;.$$
I is evaluated using the density in the parametric form (Equation 1). The latter involves the set of points $\left\{R_{i}\right\}$ at which the Gaussian density profiles are respectively centered. Averaging over the different choices of this amorphous lattice, we express the final result in terms of a site–site pair correlation function w(r) for the lattice points.
(27)$$I=-\int dr_{1}\int dr_{2}\int dRc(|r_{1}-r_{2}\left|;\hat{\varphi }\right)\phi \left(r_{1}-R\right)\phi \left(r_{2}\right)\left[\delta \left(R\right)+\rho_{0}w\left(R\right)\right]\;\;.$$
The single and double primes over $f_{\mathrm{ex}}\left(x\right)$, in the above Equation (Equation 25), respectively, denote the first and second derivatives of the function with respect to the argument *x*. For solving Equation (Equation 25), the free energy $f_{\mathrm{ex}}\left(\varphi \right)$ is taken from the standard expression of excess free energy of a hard-sphere system [39], in the form
(28)$$f_{\mathrm{ex}}\left(\varphi \right)=\frac{3}{2}\left[\frac{2\varphi -\varphi^{2}}{{\left(1-\varphi \right)}^{2}}\right]-\ln\left(1-\varphi \right)\;\;.$$
The Percus–Yevick solution for the hard sphere system [31,40] is used for the direct correlation function c(r) in the integral Equation (Equation 25).

For a fixed value of $\rho_{0}$, the total free energy (sum of the respective ideal and excess parts of the free energy) is obtained over a range of the width parameter α values by solving the MWDA equation in each specific case. Metastable amorphous states, distinct from the uniform liquid state, are identified by locating the intermediate minima of the corresponding free energy with respect to the mass localization parameter α at $\alpha =\alpha_{\min}$ for different values of the packing fraction φ. The quantity $\ell =1/\sqrt{\alpha_{\min}}$ is the localization parameter scaled with respect to σ. With increasing $\rho_{0}$ or φ, the particles are more localized, and thus the amplitudes of vibration of the particles around their respective mean position fall.

A class of minima corresponding to heterogeneous structures characterized by weak mass localization for low values of α is detected [27]. Close to freezing, these delocalized structures are more stable than the highly localized “hard-sphere glass”. However, at high densities, or packing fraction φ > 0.500, the highly localized states corresponding to large values of α (which signifying strong localization) become more stable. In Figure 10, we show the free energy minima for different φ at 0.617, 0.581, and 0.554. As the packing fraction increases, the curvatures of the free energy plots with respect to the width parameter α keeps changing. The solid-like behavior of these amorphous states with inhomogeneous density distributions is manifested in the corresponding elastic constants. The elastic constants are calculated by analyzing the nature of the local free-energy minima. We use the formulas (Equation 23) and (Equation 24) to compute the pressure and the bulk modulus by computing first and second derivatives of the free energy at the two respective minima shown in Figure 10. Using the *K* and *P*, in the Cauchy relation,
(29)$$K=\frac{5}{3}G+2\left(P-1\right)\;\;.$$
For the isotropic solid, we obtain the corresponding shear modulus *G*. These DFT results for *K*, *P*, and *G* are, respectively, shown in the main panel as well as in the two insets of Figure 11. We show the results for the two type minima depicted in Figure 9. For low packing fractions (φ < 0.580), the less localized (low α) state has higher values for the elastic constants. For high packing fractions (φ > 0.580), the corresponding elastic constants are higher for the sharply localized (high α) state. The corresponding pressure, calculated from the same DFT results, however, does not show any cross over in trend like the elastic coefficients and is always lower for the less localized state, i.e., the pressure for the less localized state is always more than that for the sharply localized state. This is shown in the inset of Figure 10. For the elastic constants, qualitative changes in relative behavior between high and low alpha minima occur with increasing packing fraction due to subtle changes in the curvatures of the corresponding free energy curves (shown in Figure 9).

Even a uniform liquid behaves like a solid over very short time scales, or equivalently in the high-frequency limit [41]. We studied this short-time elastic response of the uniform hard-sphere liquid in terms of the high-frequency elastic constants. The high-frequency elastic constants for a many-particle system with pairwise interaction are expressed in the well-known Mountain–Zwangig formulas [42,43]. However, these formulas do not go to a finite limit (at a fixed density) for the pure hard-sphere interaction.

Therefore, to calculate the high-frequency elastic constant, we use for the elastic moduli of the hard-sphere system, formulas obtained through an analysis of the stress tensor [44]. For local equilibrium, the stress tensor is expanded in terms of strains [45] assuming only instantaneous binary collision between the hard spheres. Three particle and higher-order collisions are being ignored here. From the long-wavelength expansion of the stress tensor, the corresponding shear modulus, $G_{\infty }$, and bulk modulus, $K_{\infty }$, are obtained (in units of $\rho_{0}k_{B}T$) in the form [46],
(30)$$\begin{array}{ccc} G_{\infty } & = & 1+\frac{18\varphi^{2}\left(1+\varphi \right)\left(2-\varphi \right)}{5{\left(1-\varphi \right)}^{4}}\;\;. \end{array}$$
(31)$$\begin{array}{ccc} K_{\infty } & = & \frac{5}{3}G_{\infty }+2\left[Z(\varphi )-1\right]+\frac{2}{3}\left[Z^{2}\left(\varphi \right)-1\right]\;\;. \end{array}$$
where $P/\left(\rho_{0}k_{B}T\right)=Z\left(\varphi \right)$. In reaching the above relations we have used for thermodynamics pressure *P* for the hard-sphere system the Carnahan-Starling approximation [39] as,
(32)$$Z\left(\varphi \right)=\frac{1+\varphi +\varphi^{2}-\varphi^{3}}{{\left(1-\varphi \right)}^{3}}.$$
We assume that the three quantities *K*, *G*, and *P* are related through the Cauchy linear relation (Equation 29) as was the case for the DFT model. Therefore, the bulk modulus *K* is obtained using the same linear relation. Results for these short time or high-frequency quantities are shown with dashed lines in Figure 12. We express these three thermodynamic quantities in units of $\rho_{0}k_{B}T$ where *T* is the temperature.

## 5. Discussion

In the present work, we obtained the entropy of the metastable hard-sphere liquid with inhomogeneous (coarse-grained) density, a (functional) Taylor series expansion of the free energy functional around the homogeneous state used in classical DFT models. The result from coarse-grained density functional model is matched with that from the microscopic model by using a modified distribution for the vibrational modes in the amorphous solid-like state. By choosing the modified density of states in the form of a boson peak, seen in amorphous solid-like states, we estimated the characteristic properties, like the height of the peak or position of the peak frequency. The theoretical predictions obtained from the model agree with the corresponding trends seen in experiments. With increasing density, the height $B_{h}$ of the peak decreases, while the position $\omega_{p}$ of the peak shifts to higher frequencies.

An essential characteristic of the boson peak is that it gets weaker with increasing fragility, which is related to the long-time relaxation behavior of the glass-forming material. This observed behavior follows naturally in the present model. The stronger the liquid is, the more it displays an increasing tendency to form network structures. The density profiles are more localized for the stronger glasses. Therefore the increase of fragility is synonymous with *decrease* of α, i.e., in the coarse-grained DFT models discussed here. Due to the fact that, as discussed above, the boson peak height decreases with decreasing α, it also implies a weaker boson peak for more fragile systems [47,48].

The coarse-grained model of DFT considered here can be linked to more traditional models of disordered solids built in terms of springs. The phenomena of boson peak in the disordered system are also studied in terms of a geometrically perfect crystal having random interactions between the neighbors [16]. For large values of α in the DFT model, the sharply localized density profiles are non-overlapping and are interpreted as individual harmonic oscillators having spring constant κ proportional to α [49,50]. Including fluctuations of α at different sites on the amorphous structure will be a natural extension of the present model to make it more appropriate to describe the heterogeneous glassy state.

For calculating free energy, we have used the MWDA of DFT to account for the inhomogeneous density distributions of the amorphous solid with purely hard-sphere interaction. The weighted density approximation [51], keeping up to second-order density fluctuations, accurately maps a purely repulsive hardcore system in terms of an equivalent low-density fluid. In this regard, it is useful to note that the hard-sphere solid is somewhat anomalous in the usual descriptions of lattice dynamics. In a microscopic level description, no expansion in terms of displacements from equilibrium sites exists for the Hamiltonian with purely hard-sphere interactions. Collisions entirely control the system, and between the collisions, motions of the hard spheres lose coherence very rapidly. This ballistic motion of the freely moving hard spheres in the crystal between collisions is quite analogous to the corresponding motion of the particles in the low-density fluid. In MWDA, the thermodynamic properties of the hard-sphere solid is successfully computed in terms of an equivalent liquid of much lower density.

The equivalent density $\hat{\rho }$ of the crystalline state in MWDA is generally much smaller than $\rho_{0}$, and thus for the low-density system, the PY approximation (Equation 28) for the free energy is appropriate to use. However, when MWDA is applied to describe the amorphous solid state, two qualitatively different types of minima occur, as discussed in the previous section. First, the low α minimum, the $\hat{\rho }$ comes out to be close to $\rho_{0}$, and is not small in the metastable region beyond the freezing point. Thus, the approximation (Equation 28) is not well applicable in this case. On the other hand, for the highly localized state (for large α), the $\hat{\rho }$ is much smaller than $\rho_{0}$, and, in this case, the approximation (Equation 28) is more appropriate. For the low α minimum in the MWDA results for free energy, the boson peak is identified by accounting for the new entropy in terms of a modified density of states. This was described in the previous section. For the other minimum (see Figure 9) at $\alpha =\alpha_{\min}$ on the higher side, the corresponding entropy $\hat{S}$ calculated from the partition function of the microscopic model with a Debye density of vibrational states, will agree (as shown in Figure 2) with the corresponding DFT result for the entropy obtained using the asymptotic Formula (Equation 11). This matching of entropy from the coarse-grained and microscopic models indicates that any correction for the density of vibrational modes over the Debye distribution $g_{D}\left(\omega \right)$ will imply a zero correction to entropy *S*. How this will affect the height and position of the boson peak for amorphous states with a high degree of mass localization ($\alpha_{\min}\;>>\;\alpha_{0}$), will be studied elsewhere.

## Figures and Tables

**Figure 1 entropy-23-01171-f001:**
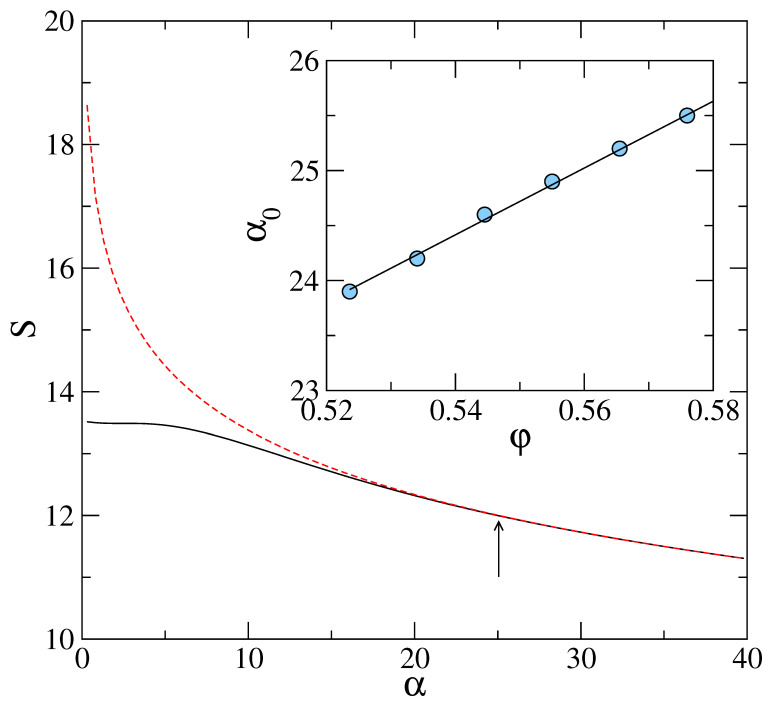
Entropy per particle *S* in units of $k_{B}$ at packing fraction φ= 0.576, obtained from the density functional result (Equation 9) (solid line) and from the asymptotic formula (Equation 11) (dashed line). Arrow marks location of $\alpha_{0}$, such that for $\alpha \;<\;\alpha_{0}$ the asymptotic formula deviates from the exact result. Inset shows $\alpha_{0}$ (in units of $\sigma^{-2}$) vs. density $\rho_{0}\sigma^{3}$ for the hard sphere system.

**Figure 2 entropy-23-01171-f002:**
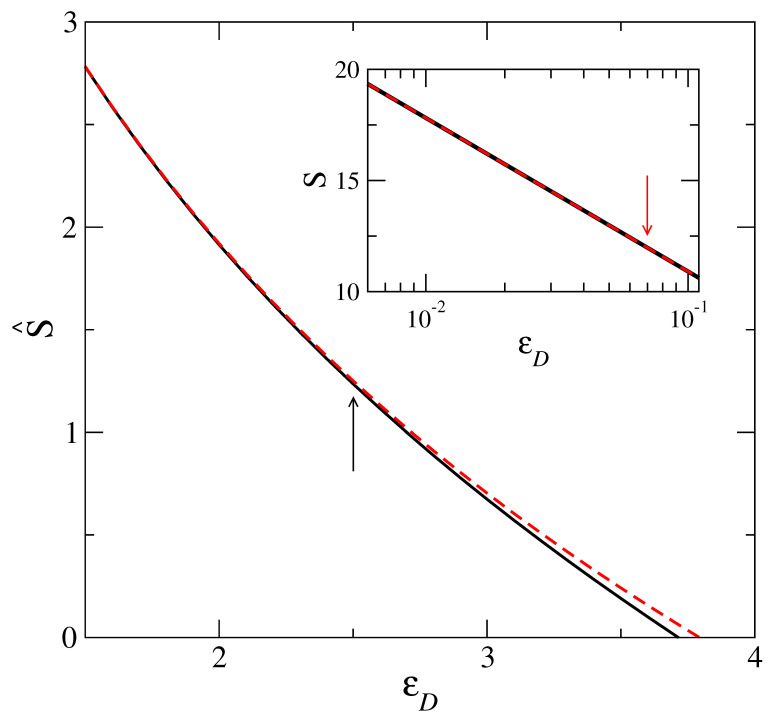
Main Panel: Comparison of the integral in right hand side of Equation (Equation 17) and the asymptotic form (Equation 20). The arrow indicates the value $\varepsilon_{D}=2.5$ at which the asymptotic result separates from the exact value. Inset focus on the part of the curve for the α range given in Figure 1 and arrow indicates the location of $\alpha_{0}$ in terms of $\varepsilon_{D}$. The width parameter α is related to $\varepsilon_{D}$ through Equation (Equation 21).

**Figure 3 entropy-23-01171-f003:**
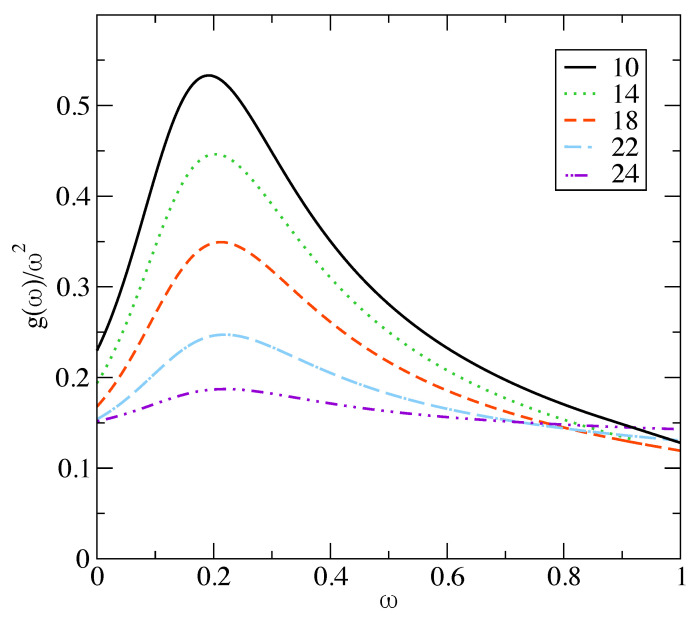
The reduced density of states $g\left(\omega \right)/\omega^{2}$ in units of $c^{3}$ (*c* is the sound speed) vs. $\omega /\omega_{D}$ ($\omega_{D}$ is the upper cutoff of Debye distribution or Debye frequency) corresponding to the width parameter α: 10 (solid), 14 (dashed), 18 (dot-dashed), 22 (dot-dashed) 24 (dot-dot-dashed) for a hard sphere system at packing fraction φ= 0.574.

**Figure 4 entropy-23-01171-f004:**
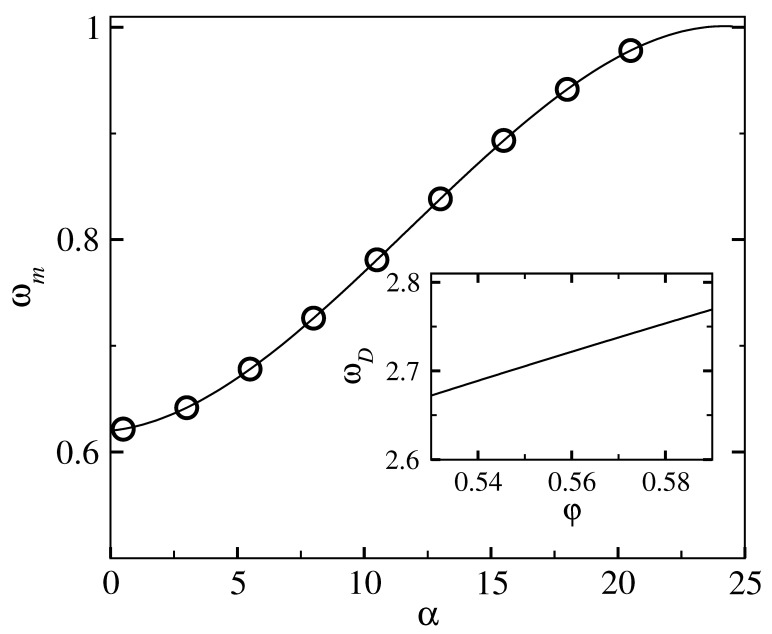
For a hard sphere system at packing fraction φ= 0.576, the upper cutoff ($\omega_{m}$) of the density of states g(ω) (modified from the Debye form) vs. $\alpha \sigma^{2}$. The frequency $\omega_{m}$ on the y axis is is scaled with the corresponding cutoff $\omega_{D}$ for the Debye distribution $g_{D}\left(\omega \right)$. The solid line is a guide to eye. Inset: $\omega_{D}$ in units of c/σ (*c* is the speed of sound) vs. packing fraction φ.

**Figure 5 entropy-23-01171-f005:**
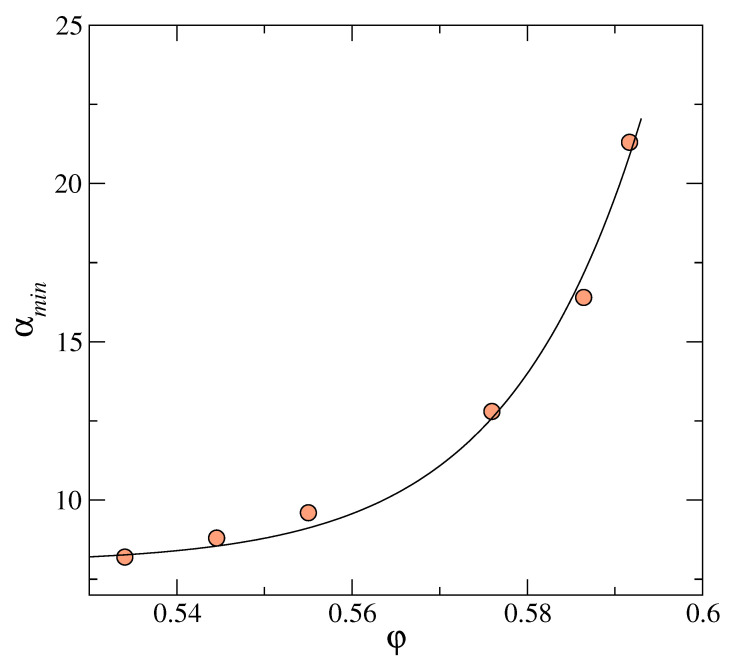
For hard-sphere system, the optimum value of the width parameter $\alpha_{\min}$ at which the total free energy has a metastable minimum vs. packing φ. The solid line is a guide to the eye.

**Figure 6 entropy-23-01171-f006:**
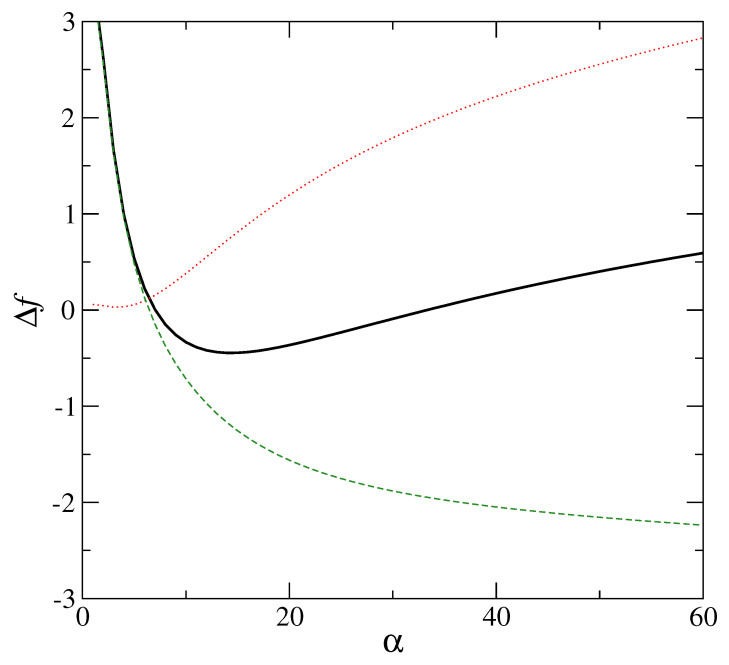
For the metastable inhomogeneous state of the hard sphere system, total free energy Δf, in excess of that of the uniform state (solid line) vs. width parameter α (in units of $\sigma^{-2}$) has a clear minimum at $\alpha \sigma^{2}=14$. The corresponding interaction part $\Delta f_{\mathrm{ex}}$ (dashed line) is calculated with density ρ(x) in Equation (Equation 1) defined in terms of Gaussian profiles centred on an *amorphous* lattice (see text); the ideal gas part $\Delta f_{\mathrm{id}}$ of the free energy difference Δf is shown as dotted line. All free energies on the Y axis are in units of $\beta^{-1}$. Packing fraction for the hard sphere system is φ= 0.576.

**Figure 7 entropy-23-01171-f007:**
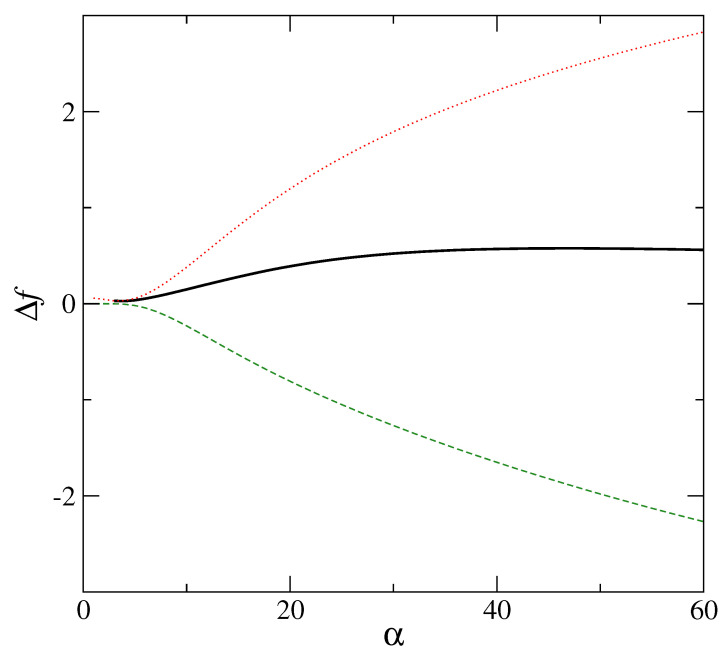
For the metastable inhomogeneous state of the hard sphere system, total free energy Δf, in excess of that of the uniform state (solid line) vs. width parameter α (in units of $\sigma^{-2}$) has no minimum in the range of α shown in Figure 6. The corresponding interaction part $\Delta f_{\mathrm{ex}}$ (dashed line) is calculated with density ρ(x) in Equation (Equation 1) defined in terms of Gaussian profiles centred on a *regular fcc* lattice with long range order; the ideal gas part $\Delta f_{\mathrm{id}}$ of the free energy difference Δf is shown as dotted line and is identical to that shown in Figure 6. All free energies on the Y axis are in units of $\beta^{-1}$. Packing fraction for the hard sphere system is φ= 0.576.

**Figure 8 entropy-23-01171-f008:**
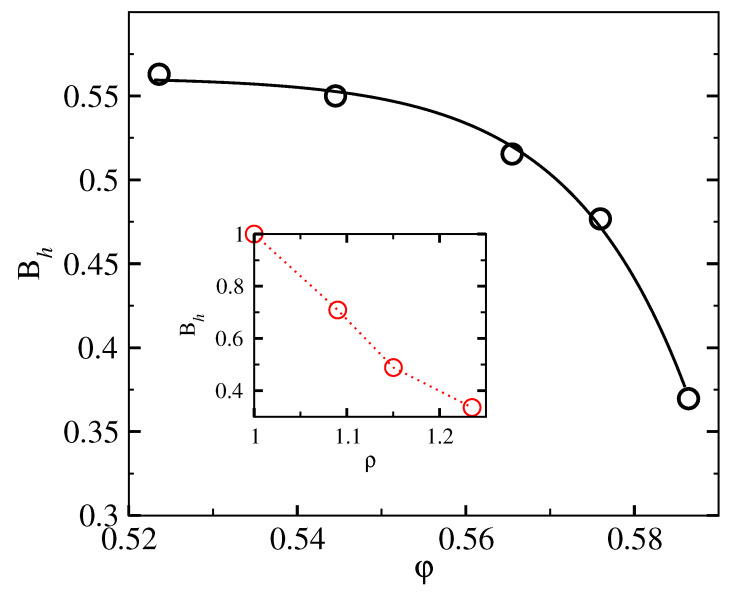
Results for properties of the modified density of states g(ω), e.g., shown in Figure 3 for the (amorphous) metastable hard sphere system with a low degree of mass localization. Main panel: boson peak height $B_{h}$ in units of $c^{3}$ (c is the sound speed) vs. packing fraction φ. Inset: experimental data [33] for boson peak height $B_{h}$ (in units of $5.5\times 10^{-4}$ meV${}^{-3}$) vs. density $\rho_{0}$ (in units of 3.66
$\mathrm{gm}/{\mathrm{cm}}^{3}$).

**Figure 9 entropy-23-01171-f009:**
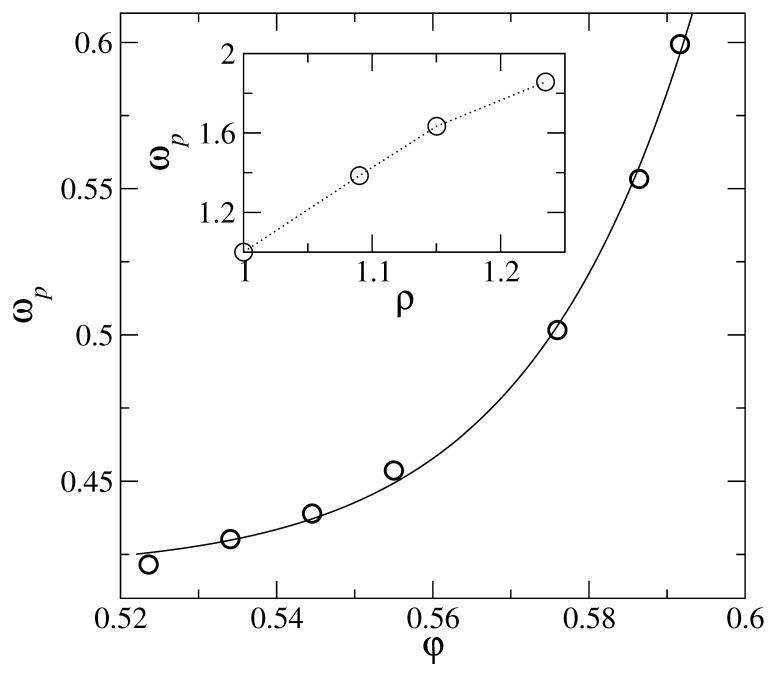
Results for properties of the modified density of states g(ω), e.g., shown in Figure 3 for the (amorphous) metastable hard sphere system with a low degree of mass localization. Main panel: boson peak frequency $\omega_{p}$ in units of c/σ (c is the sound speed) vs. packing fraction φ. Inset: experimental data [33] boson peak height $\omega_{p}$ (in units of 3.2 meV) vs. density ρ (in units of 3.66
$\mathrm{gm}/{\mathrm{cm}}^{3}$).

**Figure 10 entropy-23-01171-f010:**
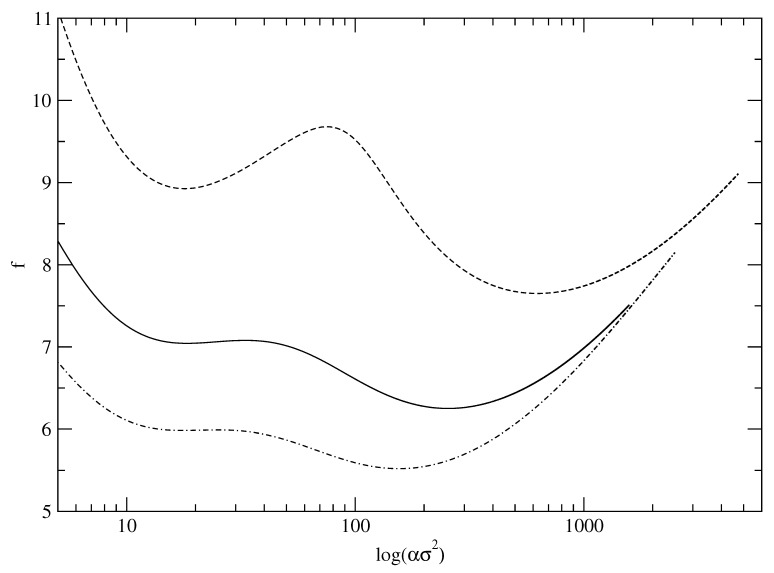
Results for free energy obtained with MWDA over a wider range of width parameter values α. The effective medium for the inhomogeneous state is calculated here with density ρ(x) (Equation (Equation 1)) defined in terms of Gaussian profiles centred on an *amorphous* lattice (see text). The free energy *f* of a hard sphere system vs. width parameter α (in units of $\sigma^{-2}$), for packing faction φ=0.617 (dashed), =0.581 (solid), and =0.554 (dot-dashed).

**Figure 11 entropy-23-01171-f011:**
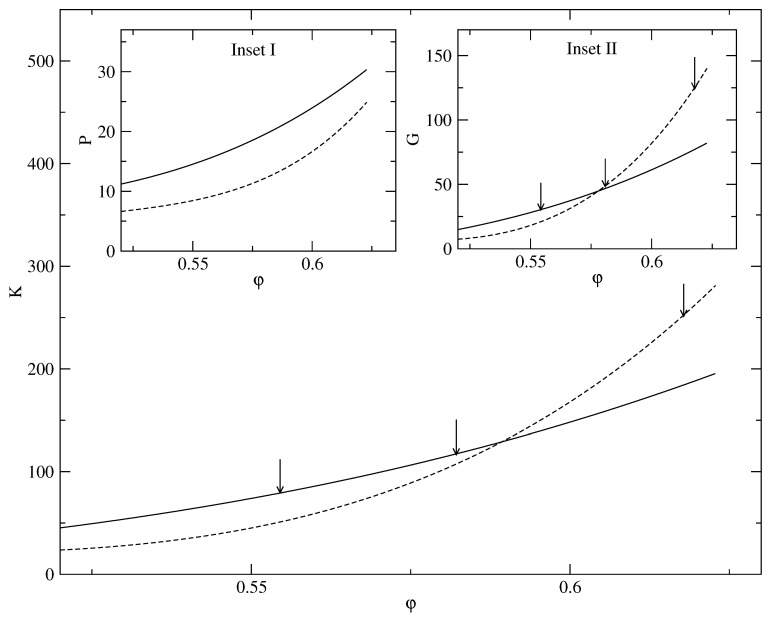
Elastic constants *K*, *G*, and pressure *P* (all three quantities in units of $\rho_{0}k_{B}T$) of the *inhomogeneous* isotropic hard sphere system vs. packing fraction φ. Results shown are obtained from respective DFT-formulas (Equation 23) for the bulk modulus *K* (main panel); (Equation 24) for Pressure (Inset I); (Equation 29) for Shear modulus *G* (Inset II). In all three respective figures solid(dashed) lines are for results obtained for the metastable state corresponding to low(high) width parameter (α) value (see text). The three vertical arrows in main panel and inset II correspond to packing fraction values, φ= 0.554, 0.581, and 0.617 for which the respective free-energy curves are shown in Figure 10.

**Figure 12 entropy-23-01171-f012:**
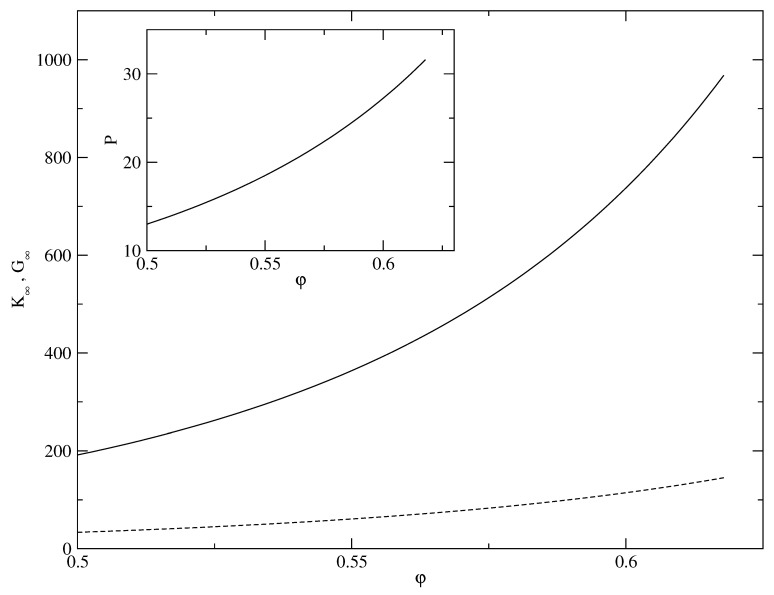
Main panel: high-frequency limit of the shear modulus $G_{\infty }$ (solid) and bulk modulus $K_{\infty }$ (dashed) for the *uniform* density hard sphere system vs. packing fraction φ. Inset: pressure *P* for the uniform hard sphere liquid vs. packing fraction φ. Here $G_{\infty }$, $K_{\infty }$, and *P* (all expressed in units of $\rho_{0}k_{B}T$) are respectively obtained [44,46] using formulas (Equation 30)–(Equation 32) in the text.

## Data Availability

The data for the plots of theoretical curves shown can be available. Not experimental data involved.

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
