# Peer review of "Localization, Disorder, and Entropy in a Coarse-Grained Model of the Amorphous Solid"

_entropy, 2021, doi:10.3390/e23091171_

Round 1
Reviewer 1 Report
The authors studied the entropy of the metastable hard-sphere liquid with inhomogeneous density with the coarse-grained description of a many-particle system used in the classical density function model. This work is of high quality and interesting to the community. I have one minor comment.
Since the α is the most important definitive parameter in this study, it would be very helpful if the author could draw a schematic phase diagram corresponding to α.
Reviewer 2 Report
The manuscript "Localization, disorder, and entropy in a coarse-grained model of the amorphous solid", by Leishangthem Premkumar, et al., seems to contain results that are interesting and sound enough to warrant publication. I have no crucial scientific objection or issue about this piece of work, but it must be said that its form should be definitely improved. In the following, I indicate the points that are a source of confusion or misunderstanding, but I encourage the authors to reread the whole manuscript.
1. The sentence "The model for disordered solid in terms of a geometrically perfect crystal having random interactions between nearest neighbors or having the particles with different masses." is ill-formulated, as it lacks the verb, and maybe also a predicate.
2. In Sec. 2.1 the authors use the grand-canonical ensemble, but then they calculate the free energy. This requires that the relation between the chemical potential and the density is inverted (as the authors correctly do), which is possible only for the ideal-gas contribution, so I encourage the authors to discuss this subtlety.
3. I was wondering whether a symbol different from m_i (which can be mistaken for a mass) can be adopted in Eq. (13).
4. The symbol w(R) in the sentence "We have obtained w(R) using the following relation:" looks like a text, not like a formula.
5. In Sec. 2.2 the authors propose an alternative (microscopic) description, shifting the focus from a collection of N atoms to a perfect gas of phonons. Maybe, few more words should be spent about the limitations or possible drawbacks of this shift.
6. Maybe the authors could spend few more words to explain how Eq. (18) is obtained.
7. The speed of sound c entering the expression of the Debye frequency is usually taken as a proper average of the speed of longitudinal and transverse modes. This point should be specified. Of course, the question about the extent to which this distinction is lost in an amorphous solid is interesting, but maybe outside the scope of the present paper.
8. In the text "being the Deby distribution" the name Debye is misspelt.
9. The sentence "The point at which the two curves (approximate and exact) as given by respective Eqns. (20) and (17), deviate marks a maximum value εmax = 2.5, is marked with an arrow ..." seems to be ill-formulated. I think that a better formulation is "The point at which the two curves (approximate and exact), as given by Eqns. (20) and (17), respectively, deviate marks a maximum value εmax = 2.5, which is marked with an arrow ...".
10. In the text "entire range of alpha value" I suggest substituting alpha with the corresponding symbol α, and maybe "values".
11. The sentence "The normalization condition (15) is for the modified g(ω) is maintained ..." seems to be ill-formulated ( ... is ... is ...). I could not grasp the authors' intention, so I urge them to reformulate this sentence.
12. I could not understand the use of "cursor" in the text "an important cursor to".
13. In Eq. (25) a hat is missing on one of the symbols η in the denominator. Likewise, the hat is systematically dropped in Eq. (28). If the authors intend to change their notation, this must be clearly stated.
14. The meaning of the symbol φ appearing in Eqs. (30) and (31) should be recalled. It should look the same in Z(φ), in the text after Eq. (31) and in Eq. (32).
15. The text "pressure Pfor" should read "pressure P for".
16. In the text "the low alpha minimum" I suggest substituting alpha with the corresponding symbol α.
Once the authors have complied with the above issues, I suggest that their manuscript may be accepted for publication.
